# Health-related quality of life and associated factors among prisoners in Gondar city prison, Northwest Ethiopia: Using structural equation modeling

**Beminate Lemma Seifu**[1]*, **Solomon Gedlu Nigatu**[2], **Lemma Derseh Gezie**[2]

**1** Department of Public Health, College of Medicine and Health Sciences, Samara University, Samara, Ethiopia, **2** Department of Epidemiology and Biostatistics, Institute of Public Health, College of Medicine and Health Sciences and Comprehensive Specialized Hospital, University of Gondar, Gondar, Ethiopia

* beminetlemma1915@gmail.com

## Abstract

### Background

Prisoners usually need more comprehensive health and social support than the general population. Due to the growing number of prisoners in Ethiopia and limited access to health service, quality of life is a key concern. Compromised health-related quality of life imposes short and long-term consequences on the prisoners, their families, and the healthcare system. In Ethiopia, there are limited studies that investigate health outcomes and health-related quality of life in this particular population. Therefore, this study aimed to assess the magnitude of health-related quality of life and associated factors among prisoners considering the multidimensional nature of health related quality of life.

### Methods

An institution-based cross-sectional study was conducted on 1,246 prisoners who were enrolled using simple random sampling. The World Health Organization Quality of Life (WHOQoL-BREF-26) and Patient Health Questionnaire (PHQ-9) tools were used to assess the HRQoL and depression among prisoners, respectively. The relationships between exogenous, mediating, and endogenous variables were identified using structural equation modeling. As the mediation of effects were present, then the direct, indirect, and total effects were determined. General fit indices of the final model were acceptable (x2/df = 1.76, $p <$ 0.001, RMSEA = 0.06, TLI = 0.90, CFI = 0.91, and SRMR = 0.06).

### Result

The mean (standard deviation) score of the overall health related quality of life was 53.25 (15.12). Having an underlying medical condition had negative total effect on health related quality of life while visits in prison had positive total effect. Having income-generating work in prison had only a direct positive effect. Whereas, older age, being married, longer duration

**Funding:** We have received the fund or financial support from the University of Gondar, Ethiopia. However, the funder had no role in the study design, data collection and analysis, decision to publish or the preparation of manuscript.

**Competing interests:** The authors have declared that no competing interests exist.

**Abbreviations:** AVE, Average Variance Extracted; CIF, Comparative Fit Index; GFI, Goodness of Fit Index; HCV, Hepatitis C Virus; HIV, Human immune deficiency virus; HRQoL, Health Related Quality of Life; IFI, Incremental Fit Index; NFI, Normated Fit Index; QoL, Quality of Life; RMSE, Root Mean Square Error; SEM, Structural Equation Modeling; TLI, Tucker Lewis Index; WHO, World Health Organization.

of imprisonment, and depression all had only a negative direct effect on one or more domains of quality of life (p<0.05).

## Conclusion

Inmates in Gondar Prison have very poor and compromised levels of physical and psychological health despite having a modest degree of overall HRQoL. The result of this study is significant for people who work in and research the prison environment because it can assist in recognizing prisoners' health needs and devising treatment procedures that take into consideration physical, psychological, environmental, and social relationship aspects.

## Background

According to World Health Organization (WHO), quality of life is defined as individuals' perceptions of their position in life in the context of the culture and value systems in which they live and in relation to their goals, expectations, standards, and concerns [1]. Health-related Quality of Life (HRQoL) is a multi-dimensional concept, which includes domains related to physical, mental, emotional, and social functioning. It goes beyond direct measures of population health, life expectancy, and causes of death, and focuses on the impact health status has on quality of life [2].

An estimated 11.5 million people have been imprisoned globally, of them about one million are found in African jails [3]. According to the 2021 United Nations Office on Drugs and Crime (UNODC) report, there are between 100,000 and 120,000 inmates in Ethiopia with approximately 4% female inmates [4].

Around 102 countries reported prison occupancy levels of over 110% [5]. In comparison to the general population, overcrowding and underfunding of prisons, as well as the misuse of imprisonment, have resulted in poor health care and sanitary conditions for prisoners, making them vulnerable to both communicable and non-communicable diseases [6–8]. Today, Ethiopian prisons are overcrowded, underfunded, and the house of a rapidly aging population. Over 112,361 incarcerated adults in Ethiopia share a disproportionate burden of infectious diseases, psychiatric, and other major medical problems [7, 9].

High prevalence of common mental disorders (CMDs) and high transmission rates for TB and HIV are commonly observed in prisons and these findings are associated with the concentration of high-risk population and with the inadequate healthcare provision in prisons due to under-resourcing and a lack of health care staff [7, 10]. Prisoners are the most vulnerable segment of the population to various types of mental disorders. The prevalence of CMDs among prisoners was found to be twice as higher as that of the general population [11–14]. In Ethiopia, the prevalence of depression, which has a direct negative impact on HRQoL among prisoners was 45.5% in Bahir Dar [15] and 55.9%, 56.4%, and 62.7% among prisoners in Mekelle, Southern Nations Nationalities and People (SNNP) and in Jimma, respectively [16–18]. The pooled prevalence of tuberculosis among prisoners in Ethiopia was high (8.33%), which in turn compromises the prisoner's HRQoL [19].

Compromised health-related quality of life have long-term significant consequences on the prisoners, their families, and the healthcare system. Lower HRQoL is linked with a higher risk of hospitalization and mortality as well as increased expenditures on the health system and families [20].

Previous studies on HRQoL of prisoners indicated that socio-demographic factors such as age, sex, marital status and educational level, co-morbid physical and mental health conditions,

and imprisonment-related characteristics such as previous imprisonment, duration of imprisonment, visit in prison and work in prison were found significantly associated with the HRQoL among prisoners [11, 21–25].

Most prisoners will eventually return to the community, bringing with them new diseases and untreated conditions that may endanger community health and add to the community's disease burden [26]. Because prisons are regulated but not closed systems, a large number of people enter, leave, and re-enter them on a regular basis. As a result, prisoners' health is an important component of public health because health issues within and outside of prisons are intertwined [27, 28].

Even though there are numerous studies conducted in Ethiopia among prisoners [17, 24, 29–32], there is a dearth of evidence about to what extent the physical, psychological, social relations, and environmental domains of HRQoL of prisoners are affected and the overall HRQoL. Furthermore, previous studies conducted out of Ethiopia among prisoners [10, 23, 33–36] have treated HRQoL as an observed variable when in reality, it is a multidimensional notion best measured using structural equation modeling. Therefore, this study aimed to assess the health-related quality of life and to explore the relationship between HRQoL and a set of individual socio-demographic, health related and imprisonment related characteristics, employing an analysis method that takes into account the complex relationships of factors that affect it.

## Methods

### Study setting, design and period

An institution based cross-sectional study was employed in central Gondar city prison among prisoners from May 2022 to June 2022. The central Gondar city prison, also known as the "Angereb prison", was built in 1930. According to data obtained from the central Gondar city prison office, approximately more than 3,117 prisoners live in the prison at the time of the study. A clinic in the prison monitors prisoners' health and has a referral link with the University of Gondar Comprehensive and Specialized Hospital (UoGCSH).

### Population, sampling procedure and sample size

The study participants were prisoners at Gondar city prison who stayed for four months and above (serving a sentence or awaiting sentence) and whose age is 18 and above. The sample size was determined by the rule of thumb for calculating the minimum adequate sample size for structural equation modeling, by the $N$:$q$ rule [37]. The ratio of the number of cases ($N$) to the number of model parameters that require statistical estimates ($q$) which ranges from 5–20 times the number of free parameters to be estimated in the hypothesized model [38]. The required sample size to address the objectives of this study was calculated by the 1:10 ratio. Considering 122 free parameters that needs to be estimated and taking a 10% non-response rate, the minimum required sample size found to be 1342. Sampling frame prepared by collecting the identification number of prisoners from the registration book. After identifying the prisoners who fulfill the inclusion criteria, study subjects was selected by simple random sampling technique using computer generated random numbers. The selected respondents participated in this study voluntarily after they read the information in the oral consent script.

### Hypothesized causal model

The present study postulates a relationship between socio-demographic characteristics, health and imprisonment-related factors as exogenous variables, the four domains of health related

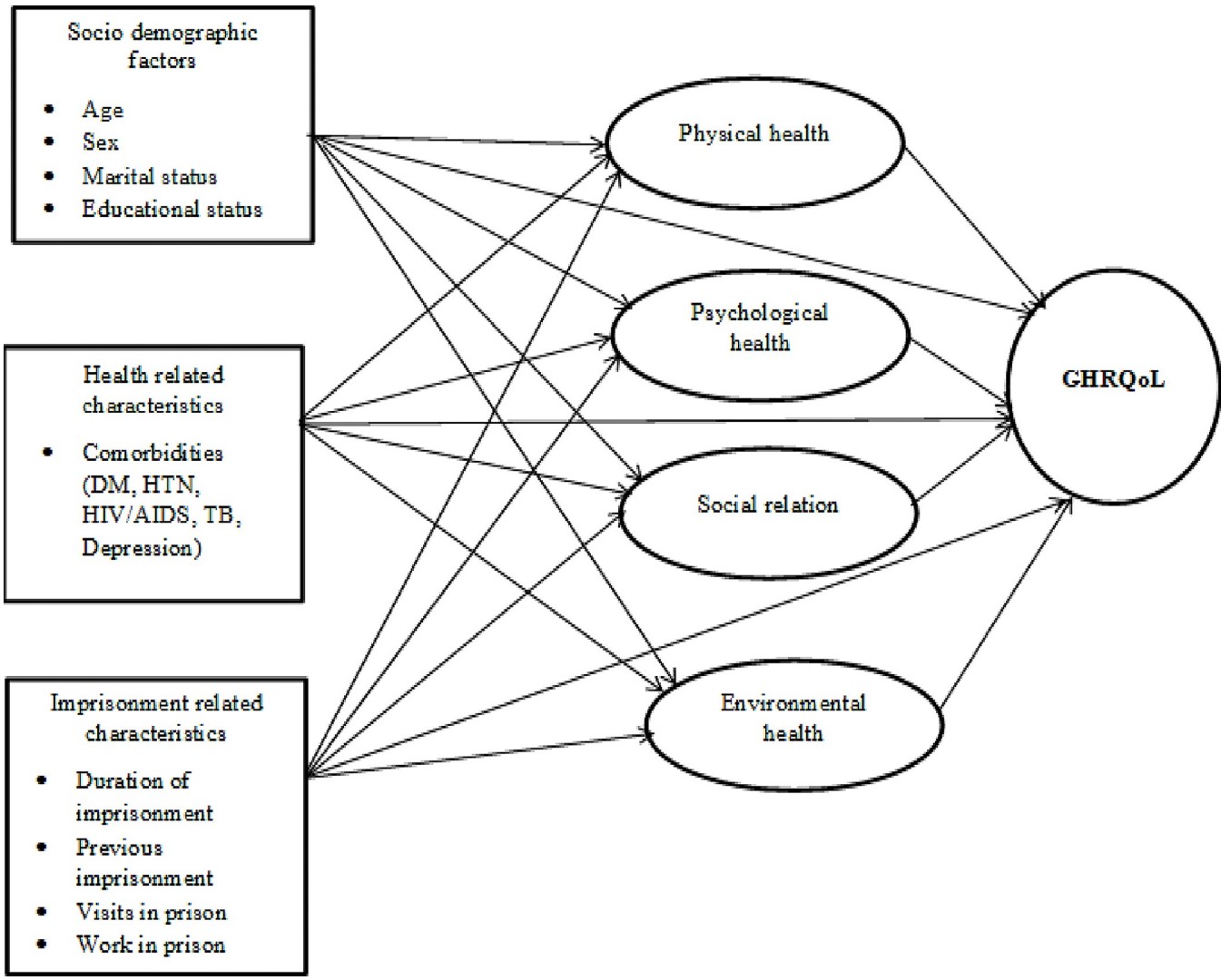

**Fig 1. Diagrammatic presentation of the hypothesized causal pathway.**

quality of life (physical, psychological, social-relations and environmental health) and global health related quality of life (GHRQoL) an endogenous variable, which contains variables related to perceived general health satisfaction and perceived quality of life. As shown in Fig 1, it posits that the socio-demographic characteristics (age, sex, marital status and educational level), Health related factors (comorbidity) and imprisonment related factors (previous imprisonment, duration of imprisonment, work in prison and visit in prison), plausibly affect the four domains of health related quality of life (HRQoL) directly, and further affect the Global Health Related Quality of Life (GHRQoL) directly or indirectly through the four domains of HRQoL (Fig 1).

## Data collection, variables, and measurements

Initially questionnaire was prepared in English version, then translated into Amharic (local language) and again translated back to English by another person to check the consistency of the meaning. Public health professionals collected the data through face-to-face interview, after receiving training on how to collect the data using both structured and standard

questionnaire tools. The tools comprised of socio-demographic characteristics, health-related (clinical and depression symptoms related) and imprisonment-related questions and the WHOQoL-BREF tools.

Health related quality of life was measured using World Health Organization QoL Instrument (WHOQoL-Brief). WHOQoL-BREF is a 26 item instrument consisting of four domains: physical health (7 items), psychological health (6 items), and the social relation (3 items), and environmental health (8 items), the overall perception of QoL and general health (2 items) (1). The score of each domain of WHOQoL-BREF were obtained by adding their respective item scores for each respondent and then the scores were transformed linearly to a 0–100 scale as suggested by WHOQoL group (1). The global HRQoL and the 4 sub-domains are latent variables with items ordered in 5-point Likert scale. Since continuous methods can be used when a variable has four or more levels [39, 40], the measurement model were analyzed considering the items as a continuous variables.

The internationally validated Patient Health Questionnaire (PHQ-9) was used to assess depression. The (PHQ-9) consists of 9 items with '0' = not at all, '1' = several days, '2' = more than half a day and '3' = nearly every day. It has a score of 0 to 27. Depression was considered using a cut point greater than or equal to five [41].

## Data processing, model building and analysis

Descriptive analysis were performed using STATA version 16, and structural examination, path analysis, and effect analysis were performed using AMOS version 24 software. Descriptive and summary statistics was reported using texts, tables and graphs. The correlation analysis of all the domains of HRQoL and the global HRQoL done using spearman's rank correlation coefficient.

Nonparametric bootstrapping using maximum likelihood estimation was used to examine the significance of effectiveness of the paths. Repetition of 500 times and 95% confidence interval were set.

In this study, to reduce the number of parameters in the analysis and maintain a reasonable degree of freedom for the model, we perform a full-item and subset-item parceling, and the items measuring each construct were averaged to create a scale score for each respondent on each of the multi-item scales. The recommended three parcels per factor by the random algorithm [42] was used for the three domains (physical, psychological and environmental health).

For the global HRQoL domain (consists overall quality of life and general health facet), latent composite (LC) or full item (all-item) parcel technique employed. Computation of this all-item-parcel approach requires a certain manipulation because specifying a latent construct with only one indicator causes under identification (i.e., $df < 0$), and under identified models cannot be computed (38). Thus, to make a model based on the all-item-parcel approach identified conventionally, the factor loading of the parcel onto the latent factor is fixed at 1.0, and the error variance ($\theta\varepsilon$) at one minus the scale's reliability coefficient (e.g., Cronbach's alpha) multiplied by the variance of the composite-score, or $\theta\varepsilon = (1-\alpha) \times s^2$. Where $\alpha$ represents the scale reliability and $s^2$ is the observed variance [42].

## Measurement model

The NC of the measurement model in this study was 1.93 (NC = x2/df, x2 = 782.50, df = 41, $p < .001$). Although it is ideal when the $P$ value of the model is not significant, it is likely to be influenced by the amount of data. Subsequently, after finding out that the overall fit indices of the measurement model in this study were satisfactory (RMSEA = 0.06, TLI = 0.94, CFI = 0.95, SRMR = 0.07), the research model was analyzed.

**Table 1. Reliability and validity of the measurement model with indices.**

| Constructs and indicators | Factor loading | Cronbach alpha | Composite reliability | AVE |
|---|---|---|---|---|
| **Physical health** | | 0.84 | 0.76 | 0.53 |
| Parce11 | 0.75 | | | |
| Parce121 | 0.61 | | | |
| Parce31 | 0.72 | | | |
| **Psychological health** | | 0.70 | 0.67 | 0.41 |
| Parc12 | 0.62 | | | |
| Parc22 | 0.61 | | | |
| Parc32 | 0.55 | | | |
| **Social-relations** | | 0.72 | 0.63 | 0.42 |
| Q20 | 0.43 | | | |
| Q22 | 0.51 | | | |
| Q28 | 0.62 | | | |
| **Environmental health** | | 0.80 | 0.76 | 0.62 |
| Parc13 | 0.68 | | | |
| Parc23 | 0.72 | | | |
| Parc33 | 0.75 | | | |

**Abbreviations:** parce11, average of Q4, Q18 and Q10; parce121, average of Q17 and Q3; parce31, average of Q16 and Q15; parce12, average of Q11 and Q19; parce22, average of Q7 and Q26; parce32, average of Q6 and Q5; parc13, average of Q25, Q23 and Q9; parc23, average of Q24, Q13 and Q12; parc33, average of Q14 and Q8.

## Test of normality and convergent validity

The test of normality was necessary to obtain accurate results from the analysis using maximum likelihood. Mardia's multivariate kurtosis was used to calculate multivariate normality. Outliers was identified by their Mahalanobis distances, which are the squared distance in standard units between the vector of an observation and the vector of sample mean for all variables. The greater the distance, the greater an observation's contribution to Mardia's multivariate kurtosis and thus to the departure from multivariate normality [43]. Unfortunately, multivariate normality was not met in this study. Therefore, maximum likelihood estimation with bootstrapping used which is an increasingly popular and promising approach in a number of contexts, and this resampling method can be used to correct fit and standard errors for non-normality in SEM [44].

The factor loading, Cronbach's alpha, composite reliability and convergent validity of each latent variables are presented in Table 1. The average variance extracted (AVE) for social-relation and psychological ranges between 0.42 and 0.41 respectively, and is below the recommended level, 0.5. The average variance extracted is a more conservative estimate of the validity of the measurement model, and based on composite reliability alone, we may conclude that the convergent validity of the construct is adequate, even though more than 50% of the variance is due to error [45]. As the composite reliability of the two constructs is well above the recommended level, the internal reliability of the measurement items is acceptable (Table 1).

## Ethical consideration

Ethical approval was obtained from the ethical review committee of the UoG College of Medicine and Health Sciences (Ref No./IPH/17/05/2022). A permission letter obtained from the central Gondar city prison. Oral consent obtained from each study subject prior to the data collection process. An oral consent script prepared and provided to the participants. However, in order to protect the participant's privacy or safety, no documentation was left behind.

Subjects who did not want to take part in the study were not forced to do so. Participants were informed that all the data obtained from them would be kept confidential by using codes instead of personal identifiers and would only be used for the purpose of the study. No incentives were given for the participants.

## Results

### Characteristics of participants

Out of 1,342 recruited prisoners, 1,246 of them answered questions, with the response rate of 92.84%. The vast majority of the participants (96.87%) were males. The median age of the sample was 27 (IQR: 22–36) years. Slightly more than half (50.32%) of inmates were not married and more than one-thirds (34.43%) attained primary education. Regarding underlying medical conditions, 311(24.96%) had an underlying comorbidity. The most commonly reported comorbidity by the respondents were tuberculosis 56 (18.01%) followed by hypertension 46 (14.79%). The majority of the prisoners 1,116 (89.57%) have never been imprisoned before, 39.41% had at least one visitor per month, and 48.31% work while in prison. At the time of the study, the median duration of imprisonment was 13 months (IQR: 5–30) (Table 2).

### The magnitude of health related quality of life

The Mean (SD) of the four domains of the WHOQoL-BREF and their summary is presented in Table 3. The mean HRQoL score of the participants found to be 53.25 (15.12). Regarding the domain scores, the HRQoL domain which achieved the lowest mean score was environmental health (33.86 (18.18)), while the highest mean score was scored for physical health domain (63.02 (19.45)) (Table 3).

### Self-rated quality of life and general health of the study participants

According to self-assessment of overall quality of life 443 (35.55%) of the participants considered their quality of life as "Very poor" and 68 (5.51%) "Very good" (Fig 2).

Regarding the general health satisfaction, 702 (56.34%) of them were "satisfied" with their general health status while 86 (6.90%) were "Very dissatisfied" (Fig 3).

### Relationships among physical, psychological, social-relationships and environmental health

The correlation analysis of all the domains of HRQoL and the global HRQoL done using spearman's rank correlation coefficient. It was found that all domains of HRQoL and the global HRQoL have had a statistically significant positive correlation with each other. Physical and global HRQoL, environmental and psychological, global HRQoL and psychological, global HRQoL and environmental health domain have a strong positive relationship (r>0.5, $p$<0.001), environmental and physical health have a moderate positive relationship (0.3< r <0.5, $p$<0.001) while the rest have a weak positive relationships (r<0.3, $p$<0.001) (Table 4).

### Factors associated with health-related quality of life among prisoners

The final model containing both structural component (relationships among latent or unobservable variables) and measurement component (relationships among latent variables and its items) is shown in Fig 4 and Table 5 respectively. In each figure all, the path coefficients were statistically significant at an alpha level of 0.05.

The final model was relatively simplify and best fitted model (x2/df = 1.76, $p$ < 0.001, RMSEA = 0.06, TLI = 0.90, CFI = 0.91, and SRMR = 0.06). Fig 4 showed significant path

**Table 2. Socio-demographic, health and imprisonment-related characteristics of prisoners in Gondar city prison, Northwest Ethiopia, from May to July 2022 (n = 1246).**

| Characteristics | Categories | | N(%) or median | IQR |
|---|---|---|---|---|
| **Sex** | Male | | 1,207 (96.87) | |
| | Female | | 39 (3.13) | |
| **Age** | | | 27 | 22–36 years |
| **Marital status** | Single | | 627 (50.32) | |
| | Married | | 479 (38.44) | |
| | Divorced | | 113 (9.07) | |
| | Widowed | | 27 (2.17) | |
| **Educational status** | No formal education | | 348 (27.93) | |
| | Primary | | 429 (34.43) | |
| | Secondary | | 360 (28.89) | |
| | Tertiary and above | | 109 (8.75) | |
| **Comorbidity** | Yes | | 311 (24.96) | |
| | No | | 935 (75.04) | |
| | Hypertension | Yes | 46 (14.79) | |
| | | No | 265 (85.21) | |
| | Diabetes mellitus | Yes | 33 (10.61) | |
| | | No | 278 (89.39) | |
| | Liver disease | Yes | 32 (10.29) | |
| | | No | 279 (89.71) | |
| | HIV/AIDS | Yes | 18 (5.79) | |
| | | No | 293 (94.21) | |
| | Tuberculosis | Yes | 56 (18.01) | |
| | | No | 255 (81.99) | |
| | Depression (n = 1246) | Yes | 863 (69.26) | |
| | | No | 383 (30.74) | |
| | Others | | 239 (76.84) | |
| **Previous imprisonment** | Yes | | 130 (10.43) | |
| | No | | 1,116 (89.57) | |
| **Duration of imprisonment** | | | 13 months | 5–30 months |
| **Visits in prison** | At least once per months | | 491 (39.41) | |
| | Few times per year | | 196 (15.73) | |
| | Never | | 559 (44.86) | |
| **Would you like to receive more visits?** | satisfied with amount of visits | | 370 (29.70) | |
| | Desires more visits | | 347 (27.85) | |
| | I desire visits | | 529 (42.46) | |
| **Work in prison** | Yes | | 602 (48.31) | |
| | No | | 644 (51.69) | |

*Note* Others = dyspepsia, hemorrhoids, headache, malaria, typhoid

coefficients of the final fitted model at a significant level of 0.05. The final model consists of 12 indicator variables (9 parcels of the 21 items), 6 observed exogenous variables, Socio-demographic characters (age and marital status), health-related characteristics (comorbidity) and, imprisonment related characteristics (duration of imprisonment, work and visits in prison). One observed endogenous variable (depression) and five unobserved endogenous variables (physical health, psychological health, social relation, environmental health and the global HRQoL) were included in the final model.

**Table 3. Magnitude of health-related quality of life of prisoners at Gondar city prison, 2022.**

| Domains of HRQoL | Observation | Mean (SD) | Min | Max | 95% CI |
|---|---|---|---|---|---|
| Physical | 1246 | 64.12 (20.99) | 0 | 100 | 62.95, 65.28 |
| Psychological | 1246 | 52.01 (18.96) | 0 | 100 | 50.95, 53.06 |
| Social-relationship | 1246 | 63.02 (19.45) | 0 | 100 | 61.94, 64.09 |
| Environmental | 1246 | 33.86 (18.18) | 0 | 100 | 32.85, 34.87 |
| Overall HRQoL | 1246 | 53.25 (15.12) | 7.14 | 98.44 | 52.41, 54.09 |

The unstandardized regression coefficients in Table 5 represented the amount of change in the dependent or mediating variables for a unit change in the independent variables/predictor. According to the indicated modification indices, error terms were inter-correlated. The standardized regression coefficients are presented in supporting information (S1 Table)

The direct effect of exogenous observed variables on the four domains of WHOQoL-BREF and on the global HRQoL; the direct effect of the four domains of WHOQoL on the global HRQoL, the direct effect of the six exogenous observed variables (marital status, comorbidity, duration of imprisonment, visit and work in prison) on depression were reported. The indirect

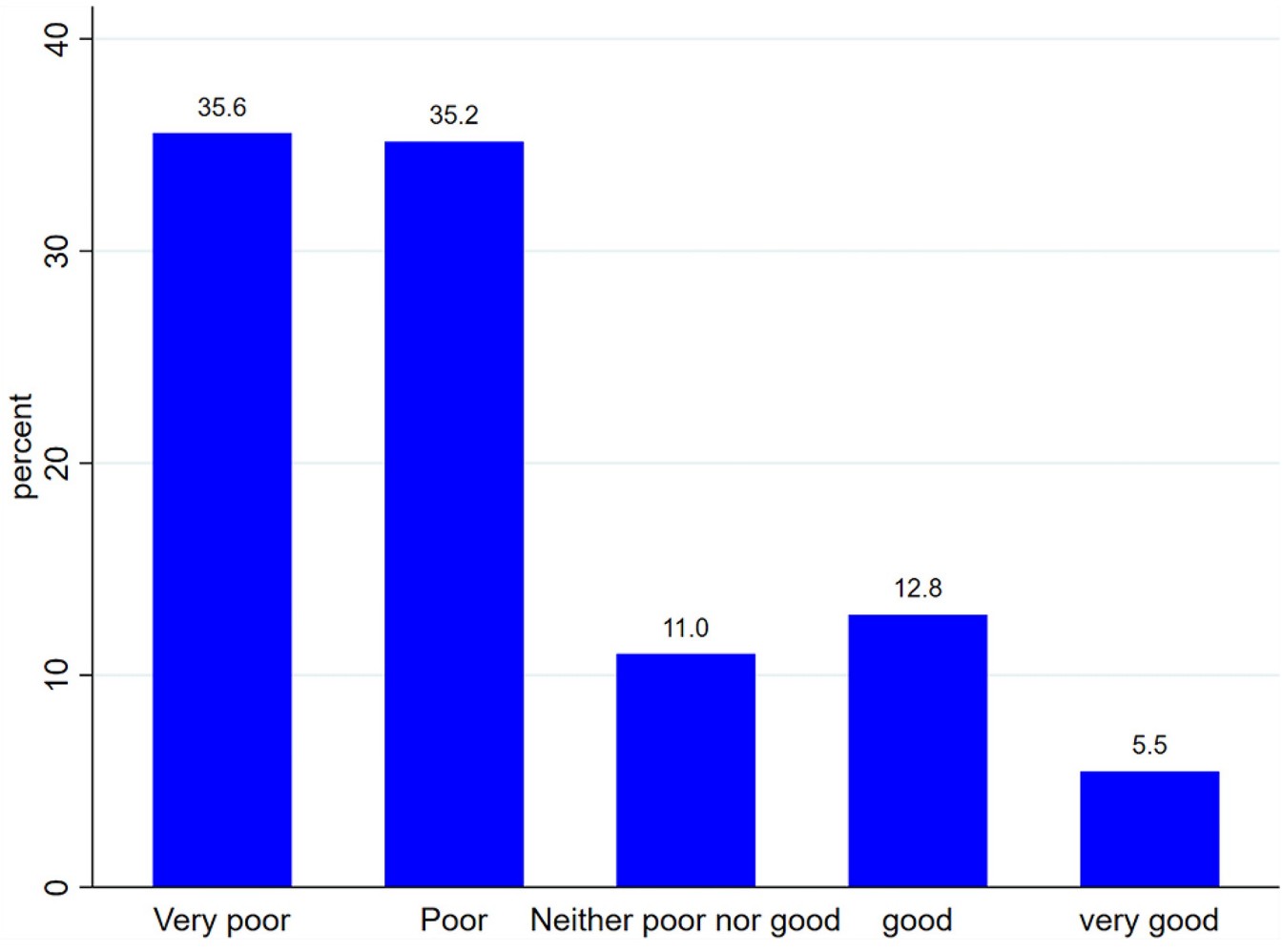

**Fig 2. Perceived self-rated QoL of prisoners at Gondar city prison, 2022.**

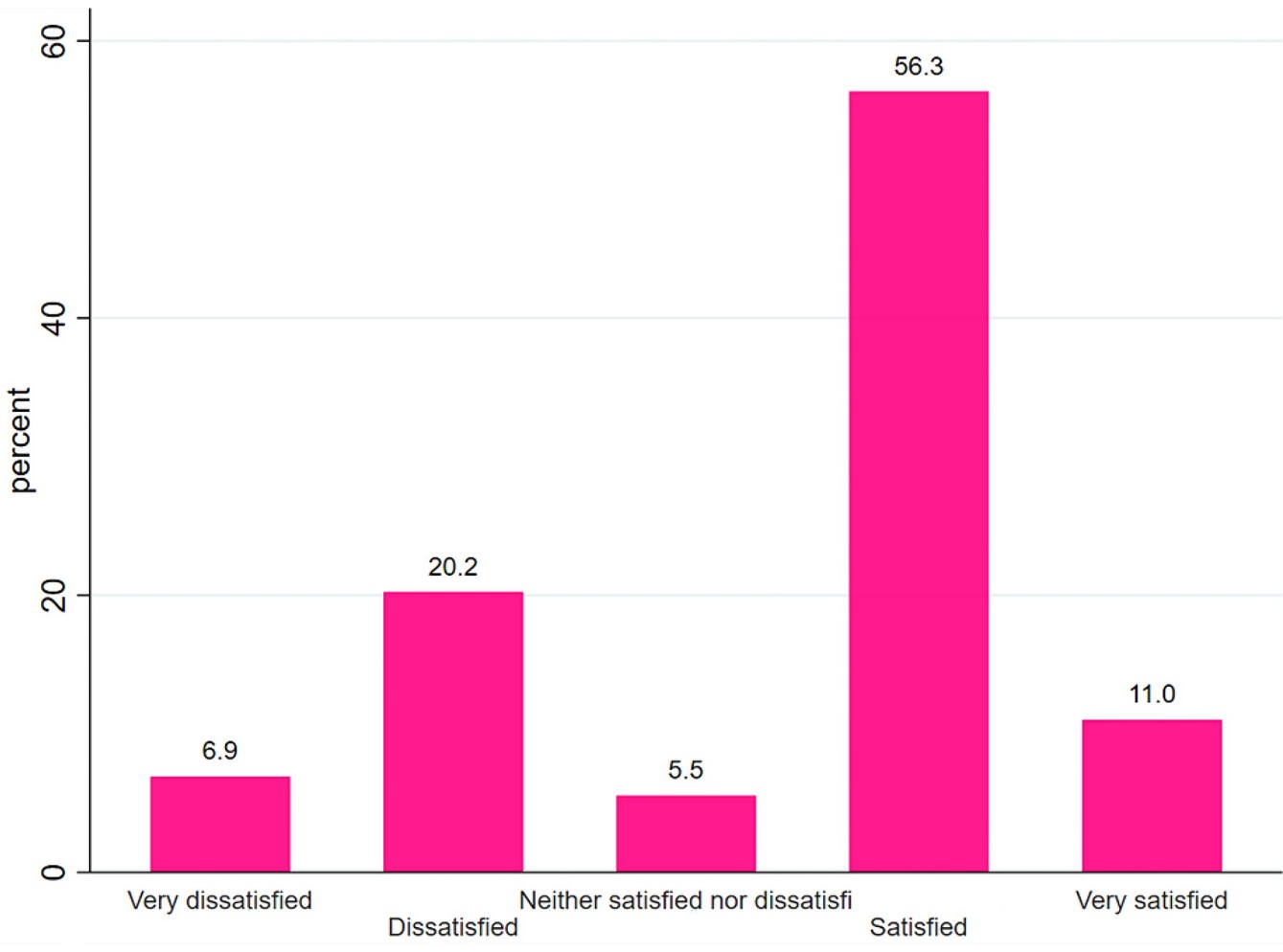

**Fig 3. Perceived self-rated General health satisfaction of prisoners at Gondar city prison, 2022.**

effect of the exogenous observed variables on the global HRQoL and its sub-domains via depression, and the direct, indirect and total effects of exogenous observed variables on the global HRQoL among prisoners are presented (Table 5 and Fig 4).

Older age had negative direct (β = -0.013, 95%CI (-0.016, -0.010)) and indirect effect (β = -0.007, 95%CI (-0.013,-0.002)) on physical health and global HRQoL domains respectively. While having a direct positive effect on environmental health (β = 0.005, 95%CI (0.002, 0.008)).

**Table 4. Spearman's rank correlation analysis of physical, psychological, social-relationships environmental and global HRQoL (n = 1246).**

|  | Physical | Psychological | Social-relationships | Environmental | Global HRQoL |
|---|---|---|---|---|---|
| **Physical** | 1 |  |  |  |  |
| **Psychological** | 0.61\*\*\* | 1 |  |  |  |
| **Social-relationships** | 0.21\*\*\* | 0.14\*\*\* | 1 |  |  |
| **Environmental** | 0.43\*\*\* | 0.61\*\*\* | 0.16\*\*\* | 1 |  |
| **Global HRQoL** | 0.64\*\*\* | 0.66\*\*\* | 0.14\*\*\* | 0.56\*\*\* | 1 |

\*\*\* $P < 0.001$

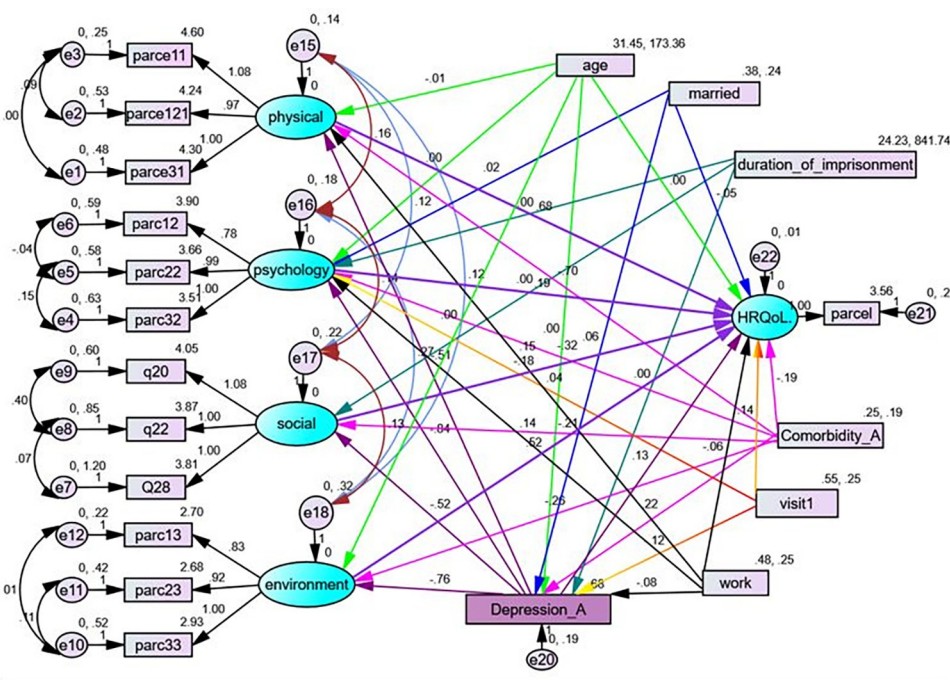

**Abbreviations:** parce11, average of Q4, Q18 and Q10. parce121, average of Q17 and Q3. parce31, average of Q16 and Q15; parce12, average of Q11 and Q19; parce22, average of Q7 and Q26; parce32, average of Q6 and Q5; parc13, average of Q25, Q23 and Q9; parc23, average of Q24, Q13 and Q12; parc33, average of Q14 and Q8, work= having work in prison and visit1=having visit in prison.

**Fig 4. SEM diagram with unstandardized coefficients displaying associations among HRQoL, socio-demographic, health related and imprisonment-related variables among prisoners in Gondar city, Northwest Ethiopia, 2022.**

Having comorbidity was both directly and indirectly related with physical, psychological, social-relation and environmental heath domains of HRQoL via the mediating variable, depression. In addition, comorbidity have a statistically significant direct negative effect (β = -0.264, 95%CI (-0.473, -0.025)) and indirect negative effect (β = -0.759, 95%CI (-1.013,-0.566)) via the four domains and depression on the global HRQoL, which result a negative total effect (β = -1.023, 95%CI (-1.170,-0.921)) on HRQoL.

Having visit in prison has a positive direct (β = 0.132, 95%CI (0.049, 0.204)), indirect (β = 0.086, 95%CI (0.046, 0.152)) and a total effect (β = 0.218, 95%CI (0.136, 0.322)) on HRQoL. Having an income generating work in prison has a direct and indirect positive effect on physical and psychological health domains with a positive total effect (β = 0.185, 95%CI (0.126, 0.243)) and (β = 0.210, 95%CI (0.141, 0.290)) respectively.

Depression had a direct negative effect on physical health (β = -0.503, 95%CI (-0.591, -0.410)), psychological health (β = -0.837, 95%CI (-0.940, -0.742)), social-relation (β = -0.563, 95%CI (-0.655, -0.461)), and environmental health (β = -0.766, 95%CI (-0.877, -0.658)) domains (Table 5).

## Discussion

In this study, the global mean score of HRQoL was 53.25 (95%CI: 52.41, 54.09), which is moderate QoL according to the WHOQoL-BREF cut-off point [46]. It could be due to, life in prison is mainly characterized by remarkable loneliness and monotony due to the physical

**Table 5. Direct, indirect and total effects of socio-demographic, health and imprisonment related factors on the four domains of WHOQoL and on global HRQoL among prisoners at Gondar city prison, Northwest Ethiopia, 2022.**

| Characteristics | Direct effect (95% CI) | Indirect effect (95% CI) | Total effect (95% CI) |
|---|---|---|---|
| **DV: Physical health** | | | |
| Age | -0.013 (-0.016–0.010)** | | |
| Marital status | | | |
| Single | - | 0 | |
| Married | - | -0.031(-0.060,-0.006)** | |
| Comorbidity | | | |
| No | 0 | 0 | 0 |
| Yes | -0.705 (-0.784, -0.619)** | -0.109 (-0.150,-0.079)** | -0.814 (-0.912, -0.721)** |
| Work in prison | | | |
| No | 0 | 0 | 0 |
| Yes | 0.145 (.094, .201) ** | 0.040 (0.015,0.069)** | 0.185(0.126,0.243)** |
| Duration of imprisonment | - | -0.002(-0.003,-0.001)** | |
| Depression | | | |
| No | 0 | | |
| Yes | -0.503 (-0.591, -0.410)** | | |
| **DV: Psychological health** | | | |
| Marital status | | | |
| Single | - | 0 | |
| Married | - | -0.051(-0.099,-0.008)** | |
| Comorbidity | | | |
| No | 0 | 0 | 0 |
| Yes | -0.315 (-0.406, -0.213)** | -0.182(-0.236,-0.139)** | -0.497(-0.608,-0.390)** |
| Duration of imprisonment | - | -0.002(-0.003,-0.001)** | - |
| Work in prison | | | |
| No | 0 | 0 | 0 |
| Yes | 0.143 (0.080, 0.203)** | 0.067(0.027,0.114)** | 0.210(0.141,0.290)** |
| Visit in prison | | | |
| No | - | 0 | |
| Yes | - | 0.103(0.062,0.147)** | |
| Depression | | | |
| No | 0 | | |
| Yes | -0.837 (-0.940, -0.742)** | | |
| **DV: Social-relationships** | | | |
| Marital status | | | |
| Single | - | 0 | |
| Married | - | -0.051 (-0.099,-0.008)** | |
| Comorbidity | | | |
| No | 0 | 0 | 0 |
| Yes | -0.222 (-0.332, -0.097)** | -0.122(-0.157,-0.091)** | -0.345(-0.456, -0.219)** |
| Duration of imprisonment | 0.003 (0.001, 0.004)** | -0.001(-0.002,-0.0001)** | 0.002(0.001,0.004)** |
| Work in prison | | | |
| No | - | 0 | |
| Yes | - | 0.045(0.018,0.082)** | |
| Visit in prison | | | |
| No | - | 0 | |
| Yes | - | 0.069(0.040,0.101)** | |

*(Continued)*

**Table 5.** (Continued)

| Characteristics | Direct effect (95% CI) | Indirect effect (95% CI) | Total effect (95% CI) |
|---|---|---|---|
| Depression | | | |
| No | 0 | | |
| Yes | -0.563 (-0.655, -0.461)** | | |
| **DV: Environmental health** | | | |
| Age | 0.005 (0.002, 0.008)** | | |
| Marital status | | | |
| Single | 0 | 0 | 0 |
| Married | - | -0.047(-0.091,-0.008)** | |
| Comorbidity | | | |
| No | 0 | 0 | 0 |
| Yes | -0.261 (-0.343, -0.162)** | -0.167(-0.228,-0.128)** | -0.428(-0.525,-0.322)** |
| Duration of imprisonment | - | -0.001(-0.002,-0.0001)** | |
| Work in prison | | | |
| No | 0 | 0 | 0 |
| Yes | | 0.061(0.025,0.106)** | |
| Visit in prison | | | |
| No | - | 0 | |
| Yes | - | 0.094(0.053,0.137)** | |
| Depression | | | |
| No | 0 | | |
| Yes | -0.766 (-0.877, -0.658)** | | |
| **DV: Depression** | | | |
| Marital status | | | |
| Single | 0 | | |
| Married | 0.061 (0.009, 0.115)** | | |
| Comorbidity | | | |
| No | 0 | | |
| Yes | 0.217 (0.163, 0.278)** | | |
| Duration of imprisonment | 0.002 (0.001, 0.003)** | | |
| Work in prison | | | |
| No | 0 | | |
| Yes | -0.080 (-0.134, -0.029)** | | |
| Visit in prison | | | |
| No | 0 | | |
| Yes | -0.123 (-0.170, -0.071)** | | |
| **DV: HRQoL** | | | |
| Physical health | 0.660 (0.305, 1.121)** | | |
| Environmental health | 0.433 (0.088, 0.777)** | | |
| Age | - | -0.007(-0.013,-0.002)** | |
| Comorbidity | | | |
| No | 0 | | |
| Yes | -0.264 (-0.473, -0.025)** | -0.759(-1.013, -0.566) | -1.023(-1.170,-0.921) |
| Duration of imprisonment | - | -0.002(-0.003, -0.001) | |
| Work in prison | | | |
| No | - | 0 | |
| Yes | - | 0.178(0.107, 0.278)** | |
| Visit in prison | | | |

(*Continued*)

**Table 5.** (Continued)

| Characteristics | Direct effect (95% CI) | Indirect effect (95% CI) | Total effect (95% CI) |
|---|---|---|---|
| No | 0 | 0 | |
| Yes | 0.132 (0.049, 0.204)** | 0.086(0.046, 0.152)** | 0.218(0.136, 0.322)** |
| Depression | | | |
| No | - | 0 | |
| Yes | - | -0.784(-0.976, -0.597)** | |

Note: *DV = dependent variable*

**p<0.05

constraint inside the few square meters of environment most often shared with others, with little or no opening toward the outside world especially countries like Ethiopia [47]. Furthermore, prisons are inherently stressful and have a non-therapeutic environment for inmates, which exacerbates prisoners' existing health problems and causes them to experience a low health related quality of life [48].

The mean score of HRQoL in our study was higher than the study conducted in Nigeria [49] using similar tool (WHOQoL BREF mean = 47.7). The discrepancy could be due to, unlike prisoners in Gondar city prison who stay the whole day time out of their cells, Nigerian prisoners can only take a walk in a courtyard for short periods while they remain inside their cell for most hours of the day with a consequent effect on their general well-being [50]. According to a study done in Belgium [51], the mean score of WHOQoL BREF was 63.18, which is greater than our finding. The possible explanation for this difference could be, unlike European countries, many African prisons including Ethiopian prisons, prisoners suffer from severe deficiencies such as high congestion, poor physical health, and sanitary conditions. Furthermore, insufficient recreational, vocational, and rehabilitation programs, and limited contact with the outside world, in turn, results in a very compromised general health and quality of life among prisoners [52].

Among the four domains of HRQoL, the highest mean score was reported in the physical health domain (mean = 64.12, 95%CI: (62.95, 65.28)) it was supported by studies reported in Belgium [51] and Greece [10]. This might be due to the majority of the study participants are young (median age = 27) so they are expected to have good physical health because health and physical fitness during young adulthood are excellent [53].

The mean score for the environmental health domain was the lowest (mean = 33.86, 95% CI: (32.85, 34.87)). This finding was lower than studies in, Belgium (mean = 61.29) [51] and France (mean = 54.72) [54]. Prison overcrowding is a major contributor to poor prison environmental conditions around the world, which could explain the low mean score of environmental health in prison. In addition to this, Overcrowding put extra strain on prisoners, potentially affecting their mental health; it may also result in deterioration of hygiene and sanitation standards, making infections and infestations more common [55].

Next to environmental health, the psychological health domain's mean score was the lowest (52.01, 95% CI (50.95, 53.06)). This is in line with studies done in Nigeria, Greece and again in Greece among prisoners who have diabetes mellitus which report the lowest mean scores for the psychological domain compared to other domains [10, 35, 49]. The poor psychological health may be related to, since being arrested the offender's identity has changed; they have lost their previous social life, and have to be separated from their family and friends; they must go through some judicial proceedings; and they usually cannot cope with their negative emotions properly in this special environment. So it's very easy for the offenders to get into a state

of severe physical and mental imbalance, leading to mal adjustment [56]. In addition to a low mean score of psychological health, the prevalence of depression in this study was high (69.26%, 95%CI (66.7%, 71.8%) compared to studies done in Addis Ababa (66.5%), Mekelle (55.9%), Hawassa (56.4%) and another evidence from the general population of Ethiopia (20.5%) [16, 17, 24, 57]. The possible explanation for prevalence of depression to be high in prison, might be due to the stressful environment of the prison, isolation from family, and a guilty feeling about their crime in prison when compared to their counterparts, the general population [16].

Age and marital status from socio-demographic factors; comorbidity from health-related characteristics; and duration of imprisonment, visit and work in prison from imprisonment-related characteristics were found to be significant predictors of health-related quality of life.

In this study, older age has a direct negative effect on the physical health domain and an indirect negative effect on global HRQoL. This is consistent with the study conducted in France [54], Greece [10], and Italy [33]. This could be due to the fact that aging (mechanical and neurological impairments affecting walking and balance, fear of falling), concomitant with the prison conditions and the restricted opportunities for movement, also play a role (refusal to take part in outside exercise due to a feeling of insecurity). These mobility problems create anxiety and depression among elderly people which in turn have a direct and indirect negative effect on the general health and overall quality of life [54]. Furthermore, older prisoners have a higher prevalence of chronic diseases, which have a negative impact on their overall well-being and quality of life.

In this study, marital status (married) found to have a statistically significant indirect negative effect on the four domains of HRQoL. This is consistent with studies done in Nigeria [50]. Married prisoners are more likely to suffer from depression, which has a direct negative impact on all domains of HRQoL because they may have a greater expectation of parental responsibility and may miss loved ones [58]. Controversially being married was positively related to the psychological and social relation domains among Chinese prisoners [56]. This disparity may explained by socioeconomic and cultural differences, differences in QoL measurement tools and methods of analysis.

According to the current study, comorbidity was found to have a direct negative effect on the four domains and a direct as well as an indirect negative effect on the global HRQoL. This is consistent with studies done in the United Kingdom (UK) [34], Australia [21], Spain [23], and Malaysia [35]. In addition, our study agrees with the study among Greece prisoners which reports, Prisoners with comorbidities had lower scores for the physical as well as the psychological aspects of health-related quality of life than prisoners without comorbidities. Also, prisoners without comorbidities had higher scores on the psychological well-being scale than prisoners with any underlying medical conditions [35]. Prisoners with health issues had worse physical, mental health, and psychological well-being outcomes because they are typically provided with a lower level of health care services than patients outside of correctional institutions [59]. In addition, the lack of medical personnel and appropriate equipment in correctional institutes discourages the procedures and complicates case management.

In the physical, psychological, social relation, and environmental health domains, depression was a statistically significant predictor of HRQoL. It also had a negative indirect effect on global HRQoL (via psychological health). This is consistent with a study done among Malaysian prisoners [21]. Being in prison can lead to low self-esteem and a negative self-image, which can exacerbate depressive symptoms. More importantly, due to a lack of trained psychiatrists and treatment facilities, early symptoms of depression are largely untreated in Ethiopia, particularly among imprisoned individuals which in turn leads to a compromised HRQoL [16].

Among imprisonment-related characteristics, having income-generating work has a statistically significant direct and indirect positive effect on HRQoL and its four domains. This is consistent with findings from Greece [10]. The possible explanation could be due to, work engagement can positively improve the quality of life of individual (i.e., improved individual and family satisfaction) and productivity, which will keep an individual mentally and physically healthy [60, 61]. Similarly, working has its own effect on reducing depression, anxiety, and related physical complaints. Furthermore, work is regarded as a source of income and social support, both of which have a positive impact on HRQoL [62].

Having a visit from outside has a positive direct and indirect effect on the global HRQoL and its domains except for physical health. This is in line with studies done in China [56] and Belgium [51]. This might be due to people who receive more support and attention from their friends or their family feel more loved, cared for, and valued, which in turn has a positive effect on their overall HRQoL [63].

Another important predictor was longer prison stay, discovered to have an indirect negative effect on overall HRQoL and its four domains. This finding is consistent with studies conducted on Portuguese [64] and England prisoners [65]. This can be explained by the fact that prison conditions are frequently inhumane and have a negative psychological impact on inmates. In contrast to our findings, a study of Norwegian prisoners discovered that time spent in prison was positively associated with perceived health and quality of life and that inmates' perceived health was better in those who had spent more time in prison [66]. This disparity can be explained by the fact that the Norwegian prison health system is well-organized, and prisoners who were in poor health before their incarceration may find help for their problems while in prison [67].

## Strengths and limitations of the study

The HRQoL was assessed using a standardized tool (WHOQoL-BREF) that has been cross-culturally validated for both developed and developing countries. In addition, structural equation modeling (SEM) was used in this study, which allows for the simultaneous analysis of the effect of multiple independent variables on several outcome variables, as well as the subsequent direct, indirect, and total evaluation of the independent variables' respective effects on the outcome variables. The following limitations should be considered when interpreting the study's findings. The data was collected through a face-to-face interview by considering the different educational levels of respondents and this may be prone to social desirability bias and could overestimate the result. Neither did this study examine the repeatability of the HRQoL evaluation due to the short study duration nor was there a matched control group drawn from the general population to allow a comparative evaluation of HRQoL in the prison population.

## Conclusion

In this study, the magnitude of health-related quality of life of prisoners was moderate. Age, marital status, comorbidity, depression, income-generating work, visitors, and duration of imprisonment found as significant factors associated with HRQoL. According to this study, having an underlying medical condition (comorbidity) had all paths coefficient significant on all the WHOQoL-BREF domains. Furthermore, the link between depression and HRQoL, as well as the relatively high prevalence of moderate to severe depression in our study population, highlight the importance of developing management strategies to improve both physical and mental health among prisoners.

## Supporting information

**S1 Table. Direct, indirect and total effects of socio-demographic, health and imprisonment related factors on the four domains of WHOQoL and on global HRQoL among prisoners at Gondar town prison, Northwest Ethiopia, 2022.**
(DOCX)

## Acknowledgments

We would like to thank the University of Gondar for giving ethical clearance to conduct this study and Gondar city prison authorities and prisoners for their kind cooperation. Furthermore, we would like to express our heartfelt gratitude and appreciation to the data collectors and supervisors for their efforts and assistance during the data collection period.

## Author Contributions

**Conceptualization:** Beminate Lemma Seifu, Solomon Gedlu Nigatu, Lemma Derseh Gezie.

**Data curation:** Beminate Lemma Seifu, Solomon Gedlu Nigatu, Lemma Derseh Gezie.

**Formal analysis:** Beminate Lemma Seifu.

**Funding acquisition:** Beminate Lemma Seifu, Solomon Gedlu Nigatu.

**Investigation:** Beminate Lemma Seifu, Lemma Derseh Gezie.

**Methodology:** Beminate Lemma Seifu, Lemma Derseh Gezie.

**Project administration:** Beminate Lemma Seifu, Solomon Gedlu Nigatu, Lemma Derseh Gezie.

**Resources:** Beminate Lemma Seifu, Solomon Gedlu Nigatu, Lemma Derseh Gezie.

**Software:** Beminate Lemma Seifu, Solomon Gedlu Nigatu.

**Supervision:** Beminate Lemma Seifu, Solomon Gedlu Nigatu, Lemma Derseh Gezie.

**Validation:** Beminate Lemma Seifu, Solomon Gedlu Nigatu, Lemma Derseh Gezie.

**Visualization:** Beminate Lemma Seifu, Solomon Gedlu Nigatu, Lemma Derseh Gezie.

**Writing – original draft:** Beminate Lemma Seifu.

**Writing – review & editing:** Beminate Lemma Seifu, Solomon Gedlu Nigatu, Lemma Derseh Gezie.

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
