## [Decision Letter · Decision Letter 0]

12 Mar 2023

PONE-D-22-30782Health-related quality of life and associated factors among prisoners in Gondar city prison, Northwest Ethiopia: using Structural Equation ModelingPLOS ONE

Dear Dr. Seifu,

Thank you for submitting your manuscript to PLOS ONE. After careful consideration, we feel that it has merit but does not fully meet PLOS ONE’s publication criteria as it currently stands. Therefore, we invite you to submit a revised version of the manuscript that addresses the points raised during the review process.

We look forward to receiving your revised manuscript.

Kind regards,

Kirubel Biruk Shiferaw, Bsc, MPH, Msc, PhD.C

Academic Editor

PLOS ONE

3. Thank you for stating the following in the Acknowledgments/ Funding Section of your manuscript:

“The data collection was financially covered by University of Gondar, Ethiopia.”

“The authors received the fund or  financial support from the university of Gondar”

“The authors received the fund or  financial support from the university of Gondar”

Reviewers' comments:

Reviewer's Responses to Questions

**Comments to the Author**

1. Is the manuscript technically sound, and do the data support the conclusions?

Reviewer #1: Yes

Reviewer #2: No

2. Has the statistical analysis been performed appropriately and rigorously? 

Reviewer #1: Yes

Reviewer #2: N/A

3. Have the authors made all data underlying the findings in their manuscript fully available?

Reviewer #1: Yes

Reviewer #2: Yes

4. Is the manuscript presented in an intelligible fashion and written in standard English?

Reviewer #1: Yes

Reviewer #2: No

5. Review Comments to the Author

Reviewer #1: The manuscript is well written and have sound structure. There are a couple of typos and grammatical errors in the text that need to be corrected e.g. line 115 page 6, line 177 page 9, line 267 page 14, line 283 page 14.

Reviewer #2: Thank you for the opportunity to review the manuscript “Health-related quality of life and associated factors among prisoners in Gondar city prison, Northwest Ethiopia: using Structural Equation Modeling”.

1: Abstract

1. the authors stated that “General fit indices of the final model were acceptable (x2/df = 1.756, p < 0.001, RMSEA = 0.058, TLI = 0.900, CFI = 0.910, and SRMR = 0.057)”. However there were not found these results in the method or result section of the manuscript. The authors supposed to report clearly these findings in the main parts of the paper.

2 .Result: the authors stated that “The mean ± standard deviation score of the overall HRQoL was 53.25±15.12”. I mean, what is the ± sign indicates that? And what is the significance of writing the standard deviation using ±?

3. In abstract, it is better to write in full word i.e., remove aberrations in your abstract section. And in the result and conclusion part of the abstract section almost the same, so in conclusion part please write to answer my question “What's implication for practice based on your results”.

2: Introduction

1.During the part of introduction, 79 references were cited, which is very rare. Some of the contents are not relevant to the object of the study, and some cited literatures are wrongly expressed, such as reference 23, 29, 38, 53 and 64. For those reports, the selected population is not focused on prisoners or quality of life.

2. The author stated “Ethiopia’s prison population rate has increased from 94 to 124 per

62 100,000 of the total population between 2000-2011 (4–6).” The data was very old and not updated. The author should be used the updated information to draw the right conclusion.

3. The author also stated that “In Ethiopia, the prevalence of depression, which has a direct negative impact on HRQoL among prisoners was 83.4% in prisons of north-west Ethiopia ……”. Some number written over there seems unreliable, so the authors should write carefully such sensitive numbers.

2. In Hypothesized causal model, what you mean by casual model? Based on the given information on the manuscript, all results was from cross sectional study. So how did you do casual model based on the observational model? Furthermore, there is no reference cited on the hypothetical model(where you develop these hypothetical model). These hypothesized model should be aso presented in your method part of the manuscript.

3: methods

1. The authors stated “The study participants were prisoners at Gondar city who stayed for four and above (serving a1sentence or awaiting sentence) and whose age is 18 and above”. In Ethiopia is there any prisoners whose age are below 18?

2. In sample size calculation the authors stated that “The required sample size to address

the objectives of this study was calculated by the 1:10 ratio”. Since the authors used rule of thumb method to calculate the sample size which is very conservative and heavily affect the power of the study. For SEM, to apply the above method the data should be multivariate normally distributed and the authors actually not consider these assumption while determining the required sample size. Obviously, most of the variable included in the study have ordinal natures, my concern is how the author mediated these controversial ideas?

I strongly suggest the authors to use other powerful statistical methods to calculated the required sample size to answer the given research questions rather than using conservative method.

3. During the sample selection the authors stated “A simple random sampling technique was used to select participants”, the authors kindly suggested to clearly describe how simple random sampling was employed in the study population?

4. The authors stated “Data related to socio-demographic and clinical factors were collected by using semi-structure and pre-tested questionnaire.” Where and when the questionary was pretested?

5. The authors used WHOQoL-BREF tools to assess the Health related quality of life the prisoners. However, the tool basically developed and validated for non-health population. Further more the tool is developed to assess the Health related quality of life patients of acute disease in two weeks and for chronic patient in one month. There for the tool is not relevant for the these study population. Therefore, the authors kindly explain why used these tool for such population

4: Statistical analysis

1. The authors suggested the analysis was done using STATA and AMOS software. What is the important of analyzing the data using two software’s?

2. The authors suggested that “In this study, to reduce the number of parameters in the analysis and maintain a reasonable degree of freedom for the model”, why the authors want to reduce the parameters? As long as the required sample size was used why not including all parameters stayed in the model?

3: Result

1. In the result section of the manuscript, the authors stated that “The correlation analysis of all the domains of HRQoL and the global HRQoL done using spearman’s rank correlation coefficient.” However, the Overall HRQoL, and its four domains are obtained by taking the average of the corresponding indicator variables and these leads to the overall HRQoL and its domains have a continuous nature. My question is why you compute spearman’s rank correlation coefficient to assess the correlation between the continues variables?

Furthermore the authors does not mention these results in the method part of the manuscript. Your results should be follow the methodology of your study.

2. The authors also stated that “The final model was relatively simplify and best fitted model.” You mean that your model is correctly estimate parameters of the model?

3. The authors stated that “The unstandardized regression coefficients in table 5 represented the amount of change in the……”. If you used unstandardized regression coefficients you are not allowed to compare the coefficients with each other’s, to do so you have to report standardized regression coefficients .

4. The authors also reports direct, indirect, and total effect of the variables, during interpreting the total effect of the variables on the outcome variables, the two variables (the one have direct effect t and indirect effect through mediating variable then have total effect) may not have the same unit of measurement. Therefore how to interpret the total effect for such scenarios?

5. Finally, the way of presentation of the result is not good, the table the author used is not readable, to increase the quality of the manuscript clearly present your results.

4: discussion

1. In the recommendation part of the manuscript some of the statements are written multiple times therefore the authors suggested to avoid writing sentences having the same idea over the manuscript.

2. Generally, the way you discuss the significant variable was completely not good. The justification the author give for the significant variable was very raw. Furthermore as I said in measurement part of the manuscript, most of your explanation were not related to what you measured(different from the tool). These makes you result unreliable and hard to comment on it.

5: Conclusion

Generally the study was done in single site the author could not have to make generalize for the whole population.

6. PLOS authors have the option to publish the peer review history of their article (what does this mean?). If published, this will include your full peer review and any attached files.

Reviewer #1: **Yes: **Professor Abdolreza Shaghaghi

Reviewer #2: No

---

## [Author Response · Author response to Decision Letter 0]

11 Apr 2023

Point-by-point response 

Point by point response for editors/reviewers comments 

PLOS ONE

Manuscript title: Health-related quality of life and associated factors among prisoners in Gondar city prison, Northwest Ethiopia: using Structural Equation Modeling 

Manuscript ID: PONE-D-22-30782

 Dear editor/reviewer. 

Dear all,

We would like to thank you for the constructive, building, and improvable comments on this manuscript that would improve the content of the manuscript. We considered each comment and clarification question of editors and reviewers on the manuscript thoroughly. Our point-by-point responses for each comment and question are described in detail on the following pages. Further, the details of changes were shown by track changes in the supplementary document attached

'Response to Reviewers

 Editor Comments:

Authors’ response: Thank you editor. We have rechecked thoroughly the manuscript and formatted as per PlOS ONE guideline. (See the revised manuscript)

Authors’ response: Thank you editor for the comment. We have addressed. 

3. Thank you for stating the following in the Acknowledgments/ Funding Section of your manuscript:

“The data collection was financially covered by University of Gondar, Ethiopia.”

“The authors received the fund or financial support from the University of Gondar”

Authors’ response: Thank you editor. We have added in the cover letter and in the enter comment section.

“The authors received the fund or financial support from the University of Gondar”

Authors’ response: Thank you editor for the comment. We have added in the cover letter (See the Cover letter)

Reviewers comment

Response to Reviewer-1

1: Abstract 

1. The authors stated that “General fit indices of the final model were acceptable (x2/df = 1.756, p < 0.001, RMSEA = 0.058, TLI = 0.900, CFI = 0.910, and SRMR = 0.057)”. However, there were not found these results in the method or result section of the manuscript. The authors supposed to report clearly these findings in the main parts of the paper. 

Authors’ response: Thank you reviewer for the comment. We have accepted the comment and included in the result section of the revised manuscript. (See result section, line number 272 and 273 & page 16) 

2. Result: the authors stated that “The mean ± standard deviation score of the overall HRQoL was 53.25±15.12”. I mean, what is the ± sign indicates that? And what is the significance of writing the standard deviation using ±?

Authors’ response: Thank you reviewer for the comment. The ± indicates the level of precision/standard error. As you can see the primary objective of this study was to determine the mean score of the overall HRQoL and therefore, we have estimated the point estimate that is mean and the level of variation (uncertainty level). Because the aim of our research is to infer/generalize our findings to the source of population which is all prisoners in Gondar prison. For this we have taken a sample of prisoners for the actual study, by applying different methods to made representative of the population but in reality no sample is the exact image of the population and there is always random error as long as we are using the sample. Therefore, to make inference we have to report the interval estimates considering the level of uncertainty. Besides, to compare our findings with previous similar studies we have to have confidence interval estimates but not based on the point estimate. These are the reasons that made us to report the interval estimates (point estimates (mean) with standard deviations). If these can’t convince you, we are very happy to remove it.

3. In abstract, it is better to write in full word i.e., remove aberrations in your abstract section. And in the result and conclusion part of the abstract section almost the same, so in conclusion part please write to answer my question “What's implication for practice based on your results”. 

Authors’ response: Thank you reviewer for the comments. We have accepted your comment and removed the abbreviations in the Abstract section. Besides, we have incorporated the implications of the findings of our study for evidence-based practice in the Abstract section. The aim of this study was to address the lack of research evidence on what the health related quality of life among Ethiopian’s prisoners looks like. In addition, this study raises a number of opportunities for future research. It is particularly important to assess the health related quality of life of prisoners in order to ensure their appropriate treatment and management. In the field of prison health, health-related quality of life is rarely researched. However, it is of particular interest to those who work in and study the prison environment because it can aid in identifying prisoners' health needs and developing therapeutic strategies that take physical, psychological, environmental, and social relational factors into account. (See Abstract section, line 45-47 and page 3)

2: Introduction

1. During the part of introduction, 79 references were cited, which is very rare. Some of the contents are not relevant to the object of the study, and some cited literatures are wrongly expressed, such as reference 23, 29, 38, 53 and 64. For those reports, the selected population is not focused on prisoners or quality of life. 

Authors’ response: Thank you so much for the comments. We have accepted the comments and cited the appropriate references. We have kept some of the reference that are not focused to the selected population in the introduction section just to show how the selected population are the most vulnerable segments of the community. Overall, we have accepted your comment and revised all the references. (See the Revised manuscript). . 

2. The author stated “Ethiopia’s prison population rate has increased from 94 to 124 per

62 100,000 of the total population between 2000-2011 (4–6).” The data was very old and not updated. The author should be used the updated information to draw the right conclusion.

Authors’ response: Thank you reviewer for raising very informative issue. We were navigating number of electronic databases to find reports about the magnitude of prisoners available in Ethiopia. However, we are unable to find recent reports about that. Besides, we were asking the responsible offices to get this data but unfortunately, we were unable to get it because they are politicizing it and keep it a secured. It was not as such easy to obtain this data that is why we have used that data.

3. The author also stated that “In Ethiopia, the prevalence of depression, which has a direct negative impact on HRQoL among prisoners was 83.4% in prisons of north-west Ethiopia ……”. Some number written over there seems unreliable, so the authors should write carefully such sensitive numbers. 

Authors’ response: Thank you reviewer for the comments. We accepted the comment and addressed it. (See the revised manuscript, line 77 and page 5)

2. In Hypothesized causal model, what you mean by casual model? Based on the given information on the manuscript, all results was from cross sectional study. So how did you do casual model based on the observational model? Furthermore, there is no reference cited on the hypothetical model (where you develop these hypothetical model). These hypothesized model should be also presented in your method part of the manuscript. 

Authors’ response: Thank you for your comments. First of all the models built in SEM generally assumes the probabilistic causality, not the deterministic causality. The deterministic causality means that given a change in a causal variable, the same consequence is observed in all cases on the outcome variable. In contrast, probabilistic causality allows changes to occur in outcomes at some probability < 1.0. Estimation of these probabilities (effects) with sample data are typically based on specific distributional assumptions, such as normality. Causality as a functional relation between two quantitative variables is preserved in this viewpoint, but causal effects are assumed to shift a probability distribution. For further though our study is observational study which was cross-sectional study, we have considered variables that pass through the basic conditions of causal inference for observational studies. For further see https://jech.bmj.com/content/76/11/960.

3: methods

1. The authors stated “The study participants were prisoners at Gondar city who stayed for four and above (serving a1sentence or awaiting sentence) and whose age is 18 and above”. In Ethiopia is there any prisoners whose age are below 18?

Authors’ response: Thank you for the concern. Yes, before the actual data collection, we have conducted a survey about the total population presented in the prison including their sex and age distribution, in our study setting we found prisoners whose age was less than 18. We have asked how this could happen and the head of the prison office assured that those who have committed serious violence could be arrested though their age is less than 18 and some of the prisoners might not know their actual age and they may respond as their age is under 18. Therefore, we have put as exclusion criteria. 

2. In sample size calculation the authors stated that “The required sample size to address

the objectives of this study was calculated by the 1:10 ratio”. Since the authors used rule of thumb method to calculate the sample size which is very conservative and heavily affect the power of the study. For SEM, to apply the above method the data should be multivariate normally distributed and the authors actually not consider these assumption while determining the required sample size. Obviously, most of the variable included in the study have ordinal natures, my concern is how the author mediated these controversial ideas? 

I strongly suggest the authors to use other powerful statistical methods to calculated the required sample size to answer the given research questions rather than using conservative method. 

Authors’ response: Thank you so much dear reviewer for raising this question. We have calculated sample size for the primary objective using mean HRQoL score reported by the previous studies using single mean population formula. Besides, we have calculated sample size for the secondary objectives for the associated factors. However, the sample size was less than 600. Then we were looking for other possibilities to have adequate sample size for the SEM model to obtain stable and reliable estimates. To the best of our knowledge, we have preferred the rule of thumb approach as we could obtained large sample size that could improve the power of the study. There are controversies in calculating sample size for SEM analysis. There are convincing evidence about the use of rule of thumb approach for SEM when the sample size obtained by other options is small. Rule of thumb method is the recommended sample size calculation for SEM because it is calculated for each parameters in the hypothesized model that needs to be estimated. The variables used to measure the outcome variable, the global HRQoL and the 4 sub-domains are latent variables with items ordered in 5-point Likert scale. Since continuous methods can be used when a variable has four or more levels, the measurement model were analyzed considering the items as a continuous variables. These are the possible reasons that forced us to use this approach to estimate the sample size. We are eager to know if you have any best way to calculate sample size for SEM.

3. During the sample selection the authors stated “A simple random sampling technique was used to select participants”, the authors kindly suggested to clearly describe how simple random sampling was employed in the study population?

Authors’ response: Thank you for the comments. We have accepted the comment and included the sampling procedure we followed to choose the samples. (See Method section, line 123-127 & page number 7)

4. The authors stated, “Data related to socio-demographic and clinical factors were collected by using semi-structure and pre-tested questionnaire.” Where and when the questionary was pretested?

Authors’ response: Thank you reviewer for the comments. We conducted a pretest before the actual data collection, it was a month before. Which was conducted among prisoners in Bahr Dar city prison as these prisons have similar baseline characteristics with respect to factors that could influence HRQoL of prisoners. 

5. The authors used WHOQoL-BREF tools to assess the Health related quality of life the prisoners. However, the tool basically developed and validated for non-health population. Further more the tool is developed to assess the Health related quality of life patients of acute disease in two weeks and for chronic patient in one month. There for the tool is not relevant for the these study population. Therefore, the authors kindly explain why used these tool for such population

Authors’ response: Thank you reviewer for the comment. We have used the WHOQoL-BREF for this study due to the generic and comprehensive character of the instrument, the good psychometric properties in different populations and its international applicability as a WHO instrument. This tool has validated in different population groups including prison population, it was used by previous researchers as the tool is more comprehensive and better than the other psychometric tools that measure health related quality of life. Like any psychometric tool, this tool has its own limitation such as the instrument is subjective though it is in line with the WHO definitions of quality of life, and we have acknowledged the limitations of the study. The WHOQoL-BREF tool has been validated in several prison populations including Dutch, Irish, French, and Nigerian prisoners. In addition, we have considered the domains as a latent variable because this tool may not capture the domains exactly. 

4: Statistical analysis

1. The authors suggested the analysis was done using STATA and AMOS software. What is the important of analyzing the data using two software’s?

Authors’ response: Thank you reviewer for the comment. We used these software for different purposes. We used STATA for data management because AMOS do not have this feature, and we used AMOS to run structural equation model because AMOS is specially used for performing structural equation modelling, path analysis and confirmatory factor analysis (CFA).

2. The authors suggested that “In this study, to reduce the number of parameters in the analysis and maintain a reasonable degree of freedom for the model”, why the authors want to reduce the parameters? As long as the required sample size was used why not including all parameters stayed in the model? 

Authors’ response: Thank you reviewer for the comment. To reduce the number of parameters was not to mean to reduce the parameters by putting them out of the model, what we did was aggregating (parceling) the related items for psychometric and modeling benefits. It is a pre-modeling method to obtain fewer and more reliable indicators of constructs for use with latent variable models. In this study, to reduce the number of parameters in the analysis and maintain a reasonable degree of freedom for the model, we perform a full-item and subset-item parceling. Parceling has psychometrics and modeling merits. Enhancement of scale communality, Reconstructing and normalizing the construct distribution and increases the given model’s efficiency to define the latent construct are some of psychometric merits of parceling. Modeling-Related advantages of Parceling is Estimation stability. Item-based solutions are often unstable and take more iterations to converge, yielding relatively large standard errors of the measurement-level parameters. One consequence stemming from these problems is estimation instability. That is, parameters estimated in item-based solutions may vary substantially even when only small changes are made to the model, making it difficult for researchers to generalize findings. The other aim was to obtain a relatively simple or parsimonious model. 

3. Result

1. In the result section of the manuscript, the authors stated that “The correlation analysis of all the domains of HRQoL and the global HRQoL done using spearman’s rank correlation coefficient.” However, the Overall HRQoL, and its four domains are obtained by taking the average of the corresponding indicator variables and these leads to the overall HRQoL and its domains have a continuous nature. My question is why you compute spearman’s rank correlation coefficient to assess the correlation between the continues variables?

Furthermore the authors does not mention these results in the method part of the manuscript. Your results should be follow the methodology of your study. 

Authors’ response: Thank you for the concern. As you know the Spearman’s rank correlation is be used in two specific scenarios: when working with ranked data and when one or more extreme outliers are present. It measures the strength and direction of association between two ranked variables and as you know WHOQoL items has ordinal nature (Since continuous methods can be used when a variable has four or more levels, the measurement model were analyzed considering the items as a continuous variables). We used spearman’s rank correlation coefficient due to the presence of outliers in our working data because when extreme outliers are present in a dataset, Pearson’s correlation coefficient is highly affected. Therefore, when there is extreme values the spearman correlation coefficient is appropriate that is why we used in our case.

2. The authors also stated, “The final model was relatively simplify and best fitted model.” You mean that your model is correctly estimate parameters of the model?

Authors’ response: Thank you for the comment. It is to mean the model is parsimonious, better fits the data. In statistical modelling, there is no correct model, all models are wrong but we have to find the model that approximate to the truth. Therefore, in our study we have fitted a model with different parameter combination and structure, and compared them with model fit indices. Finally, all the model fit indices of the reported model were satisfactory 

3. The authors stated that “The unstandardized regression coefficients in table 5 represented the amount of change in the……”. If you used unstandardized regression coefficients you are not allowed to compare the coefficients with each other’s, to do so you have to report standardized regression coefficients .

Authors’ response: Thank you reviewer for the comment. We have addressed your comment in the supporting information. We have reported both standardized and unstandardized coefficients. In SEM, it is often advised to report both unstandardized and standardized coefficients, because they present different and mutually exclusive information. Unstandardized coefficients contain information about both the variance and the mean, and thus are essential for prediction. (See the revised manuscript)

4. The authors also reports direct, indirect, and total effect of the variables, during interpreting the total effect of the variables on the outcome variables, the two variables (the one have direct effect t and indirect effect through mediating variable then have total effect) may not have the same unit of measurement. Therefore how to interpret the total effect for such scenarios?

Authors’ response: Thank you reviewer for the concern. In this study, we have considered depression and the four domains of HRQoL as mediator variables and therefore, a variable can have direct, indirect through mediating variables and total effects then we found coefficients for each effects of the variables. Then we have interpreted the coefficients based on their direction and magnitude of association. So the interpretation is straight forward by looking the magnitude and the sign of the coefficients because the unit of measurements were the same. 

5. Finally, the way of presentation of the result is not good, the table the author used is not readable, and to increase the quality of the manuscript clearly present your results.

Authors’ response: Thank you for the comments. We have modified based on your comments, please see the revised manuscript. 

4: Discussion

1. In the recommendation part of the manuscript some of the statements are written multiple times therefore the authors suggested to avoid writing sentences having the same idea over the manuscript. 

Authors’ response: Thank you reviewer for the comment. We have addressed it.

2. Generally, the way you discuss the significant variable was completely not good. The justification the author give for the significant variable was very raw. Furthermore as I said in measurement part of the manuscript, most of your explanation were not related to what you measured(different from the tool). These makes you result unreliable and hard to comment on it. 

Authors’ response: Thank you reviewer for the comment. We have addressed them.

5: Conclusion 

Generally, the study was done in single site the author could not have to make generalize for the whole population. 

Authors’ response: We thank you for the comment. You are right we cannot extrapolate this results to the whole population because we have conducted this research among prisoners and therefore we can generalize to all prisoners in the study area but we can’t generalize for prisoners out of the study area.

---

## [Decision Letter · Decision Letter 1]

11 Jul 2023

PONE-D-22-30782R1Health-related quality of life and associated factors among prisoners in Gondar city prison, Northwest Ethiopia: using Structural Equation ModelingPLOS ONE

Dear Dr. Seifu,

Thank you for submitting your manuscript to PLOS ONE. After careful consideration, we feel that it has merit but does not fully meet PLOS ONE’s publication criteria as it currently stands. Therefore, we invite you to submit a revised version of the manuscript that addresses the points raised during the review process.

The manuscript has been revised and most of the reviewer's concerns have been addressed. The manuscript is well written. In relation to the items raised by the reviewer, please address the following points before more consideration:

- The ± sign was not used anymore to describe SD or SE and you could present the level of variation between parentheses.  

- Please present  Ethiopia as a keyword.

- Please summarize the background section of the abstract.

- Use the same number of decimals throughout the text.

- Present "DV: Physical health" in Table 5 in legend.

- Instead of a null and general sentence at the beginning of the discussion section as "This study investigated the HRQoL in a prison population and explored the relationship between HRQoL and a set of individual socio-demographic, health-related characteristics and  characteristics of detention." present your result.

We look forward to receiving your revised manuscript.

Kind regards,

Kamal Gholipour, PhD

Academic Editor

PLOS ONE

Journal Requirements:

Reviewers' comments:

Reviewer's Responses to Questions

**Comments to the Author**

1. If the authors have adequately addressed your comments raised in a previous round of review and you feel that this manuscript is now acceptable for publication, you may indicate that here to bypass the “Comments to the Author” section, enter your conflict of interest statement in the “Confidential to Editor” section, and submit your "Accept" recommendation.

Reviewer #1: All comments have been addressed

Reviewer #2: (No Response)

2. Is the manuscript technically sound, and do the data support the conclusions?

Reviewer #1: Yes

Reviewer #2: No

3. Has the statistical analysis been performed appropriately and rigorously? 

Reviewer #1: Yes

Reviewer #2: No

4. Have the authors made all data underlying the findings in their manuscript fully available?

Reviewer #1: Yes

Reviewer #2: Yes

5. Is the manuscript presented in an intelligible fashion and written in standard English?

Reviewer #1: Yes

Reviewer #2: No

6. Review Comments to the Author

Reviewer #1: The authors' amendments in this revised version according to the reviewers' comments are sound and acceptable. No further correction is required.

Reviewer #2: The authors did not address my previous comments and questions. Therefore, I can't move further with the same comments anymore.

7. PLOS authors have the option to publish the peer review history of their article (what does this mean?). If published, this will include your full peer review and any attached files.

Reviewer #1: **Yes: **Professor Abdolreza Shaghaghi

Reviewer #2: No

---

## [Author Response · Author response to Decision Letter 1]

24 Jul 2023

Response to Reviewer-2

1: Abstract 

1. The authors stated that “General fit indices of the final model were acceptable (x2/df = 1.756, p < 0.001, RMSEA = 0.058, TLI = 0.900, CFI = 0.910, and SRMR = 0.057)”. However, there were not found these results in the method or result section of the manuscript. The authors supposed to report clearly these findings in the main parts of the paper. 

Authors’ response: Thank you reviewer for the comment. We have accepted the comment and included in the result section of the revised manuscript. 

2. Result: the authors stated that “The mean ± standard deviation score of the overall HRQoL was 53.25±15.12”. I mean, what is the ± sign indicates that? And what is the significance of writing the standard deviation using ±?

Authors’ response: Thank you reviewer for the comment. The ± indicates the level of precision/standard error. As you can see the primary objective of this study was to determine the mean score of the overall HRQoL and therefore, we have estimated the point estimate that is mean and the level of variation (uncertainty level). Because the aim of our research is to infer/generalize our findings to the source of population which is all prisoners in Gondar prison. For this we have taken a sample of prisoners for the actual study, by applying different methods to made representative of the population but in reality no sample is the exact image of the population and there is always random error as long as we are using the sample. Therefore, to make inference we have to report the interval estimates considering the level of uncertainty. Besides, to compare our findings with previous similar studies we have to have confidence interval estimates but not based on the point estimate. These are the reasons that made us to report the interval estimates (point estimates (mean) with standard deviations). If these can’t convince you, we are very happy to remove it.

3. In abstract, it is better to write in full word i.e., remove aberrations in your abstract section. And in the result and conclusion part of the abstract section almost the same, so in conclusion part please write to answer my question “What's implication for practice based on your results”. 

Authors’ response: Thank you reviewer for the comments. We have accepted your comment and removed the abbreviations in the Abstract section. Besides, we have incorporated the implications of the findings of our study for evidence-based practice in the Abstract section. The aim of this study was to address the lack of research evidence on what the health related quality of life among Ethiopian’s prisoners looks like. In addition, this study raises a number of opportunities for future research. It is particularly important to assess the health related quality of life of prisoners in order to ensure their appropriate treatment and management. In the field of prison health, health-related quality of life is rarely researched. However, it is of particular interest to those who work in and study the prison environment because it can aid in identifying prisoners' health needs and developing therapeutic strategies that take physical, psychological, environmental, and social relational factors into account. (See Abstract section, line 45-47 and page 3)

2: Introduction

1. During the part of introduction, 79 references were cited, which is very rare. Some of the contents are not relevant to the object of the study, and some cited literatures are wrongly expressed, such as reference 23, 29, 38, 53 and 64. For those reports, the selected population is not focused on prisoners or quality of life. 

Authors’ response: Thank you so much for the comments. We have accepted the comments and cited the appropriate references. We have kept some of the reference that are not focused to the selected population in the introduction section just to show how the selected population are the most vulnerable segments of the community. Overall, we have accepted your comment and revised all the references. (See the Revised manuscript). . 

2. The author stated “Ethiopia’s prison population rate has increased from 94 to 124 per

62 100,000 of the total population between 2000-2011 (4–6).” The data was very old and not updated. The author should be used the updated information to draw the right conclusion.

Authors’ response: Thank you reviewer for raising very informative issue. We were navigating number of electronic databases to find reports about the magnitude of prisoners available in Ethiopia. However, we are unable to find recent reports about that. Besides, we were asking the responsible offices to get this data but unfortunately, we were unable to get it because they are politicizing it and keep it a secured. It was not as such easy to obtain this data that is why we have used that data.

3. The author also stated that “In Ethiopia, the prevalence of depression, which has a direct negative impact on HRQoL among prisoners was 83.4% in prisons of north-west Ethiopia ……”. Some number written over there seems unreliable, so the authors should write carefully such sensitive numbers. 

Authors’ response: Thank you reviewer for the comments. We accepted the comment and addressed it. (See the revised manuscript, line 77 and page 5)

2. In Hypothesized causal model, what you mean by casual model? Based on the given information on the manuscript, all results was from cross sectional study. So how did you do casual model based on the observational model? Furthermore, there is no reference cited on the hypothetical model (where you develop these hypothetical model). These hypothesized model should be also presented in your method part of the manuscript. 

Authors’ response: Thank you for your comments. First of all the models built in SEM generally assumes the probabilistic causality, not the deterministic causality. The deterministic causality means that given a change in a causal variable, the same consequence is observed in all cases on the outcome variable. In contrast, probabilistic causality allows changes to occur in outcomes at some probability < 1.0. Estimation of these probabilities (effects) with sample data are typically based on specific distributional assumptions, such as normality. Causality as a functional relation between two quantitative variables is preserved in this viewpoint, but causal effects are assumed to shift a probability distribution. For further though our study is observational study which was cross-sectional study, we have considered variables that pass through the basic conditions of causal inference for observational studies. For further see https://jech.bmj.com/content/76/11/960.

3: methods

1. The authors stated “The study participants were prisoners at Gondar city who stayed for four and above (serving a1sentence or awaiting sentence) and whose age is 18 and above”. In Ethiopia is there any prisoners whose age are below 18?

Authors’ response: Thank you for the concern. Yes, before the actual data collection, we have conducted a survey about the total population presented in the prison including their sex and age distribution, in our study setting we found prisoners whose age was less than 18. We have asked how this could happen and the head of the prison office assured that those who have committed serious violence could be arrested though their age is less than 18 and some of the prisoners might not know their actual age and they may respond as their age is under 18. Therefore, we have put as exclusion criteria. 

2. In sample size calculation the authors stated that “The required sample size to address

the objectives of this study was calculated by the 1:10 ratio”. Since the authors used rule of thumb method to calculate the sample size which is very conservative and heavily affect the power of the study. For SEM, to apply the above method the data should be multivariate normally distributed and the authors actually not consider these assumption while determining the required sample size. Obviously, most of the variable included in the study have ordinal natures, my concern is how the author mediated these controversial ideas? 

I strongly suggest the authors to use other powerful statistical methods to calculated the required sample size to answer the given research questions rather than using conservative method. 

Authors’ response: Thank you so much dear reviewer for raising this question. We have calculated sample size for the primary objective using mean HRQoL score reported by the previous studies using single mean population formula. Besides, we have calculated sample size for the secondary objectives for the associated factors. However, the sample size was less than 600. Then we were looking for other possibilities to have adequate sample size for the SEM model to obtain stable and reliable estimates. To the best of our knowledge, we have preferred the rule of thumb approach as we could obtained large sample size that could improve the power of the study. There are controversies in calculating sample size for SEM analysis. There are convincing evidence about the use of rule of thumb approach for SEM when the sample size obtained by other options is small. Rule of thumb method is the recommended sample size calculation for SEM because it is calculated for each parameters in the hypothesized model that needs to be estimated. The variables used to measure the outcome variable, the global HRQoL and the 4 sub-domains are latent variables with items ordered in 5-point Likert scale. Since continuous methods can be used when a variable has four or more levels, the measurement model were analyzed considering the items as a continuous variables. These are the possible reasons that forced us to use this approach to estimate the sample size. We are eager to know if you have any best way to calculate sample size for SEM.

3. During the sample selection the authors stated “A simple random sampling technique was used to select participants”, the authors kindly suggested to clearly describe how simple random sampling was employed in the study population?

Authors’ response: Thank you for the comments. We have accepted the comment and included the sampling procedure we followed to choose the samples. (See Method section, line 123-127 & page number 7)

4. The authors stated, “Data related to socio-demographic and clinical factors were collected by using semi-structure and pre-tested questionnaire.” Where and when the questionary was pretested?

Authors’ response: Thank you reviewer for the comments. We conducted a pretest before the actual data collection, it was a month before. Which was conducted among prisoners in Bahr Dar city prison as these prisons have similar baseline characteristics with respect to factors that could influence HRQoL of prisoners. 

5. The authors used WHOQoL-BREF tools to assess the Health related quality of life the prisoners. However, the tool basically developed and validated for non-health population. Further more the tool is developed to assess the Health related quality of life patients of acute disease in two weeks and for chronic patient in one month. There for the tool is not relevant for the these study population. Therefore, the authors kindly explain why used these tool for such population

Authors’ response: Thank you reviewer for the comment. We have used the WHOQoL-BREF for this study due to the generic and comprehensive character of the instrument, the good psychometric properties in different populations and its international applicability as a WHO instrument. This tool has validated in different population groups including prison population, it was used by previous researchers as the tool is more comprehensive and better than the other psychometric tools that measure health related quality of life. Like any psychometric tool, this tool has its own limitation such as the instrument is subjective though it is in line with the WHO definitions of quality of life, and we have acknowledged the limitations of the study. The WHOQoL-BREF tool has been validated in several prison populations including Dutch, Irish, French, and Nigerian prisoners. In addition, we have considered the domains as a latent variable because this tool may not capture the domains exactly. 

4: Statistical analysis

1. The authors suggested the analysis was done using STATA and AMOS software. What is the important of analyzing the data using two software’s?

Authors’ response: Thank you reviewer for the comment. We used these software for different purposes. We used STATA for data management because AMOS do not have this feature, and we used AMOS to run structural equation model because AMOS is specially used for performing structural equation modelling, path analysis and confirmatory factor analysis (CFA).

2. The authors suggested that “In this study, to reduce the number of parameters in the analysis and maintain a reasonable degree of freedom for the model”, why the authors want to reduce the parameters? As long as the required sample size was used why not including all parameters stayed in the model? 

Authors’ response: Thank you reviewer for the comment. To reduce the number of parameters was not to mean to reduce the parameters by putting them out of the model, what we did was aggregating (parceling) the related items for psychometric and modeling benefits. It is a pre-modeling method to obtain fewer and more reliable indicators of constructs for use with latent variable models. In this study, to reduce the number of parameters in the analysis and maintain a reasonable degree of freedom for the model, we perform a full-item and subset-item parceling. Parceling has psychometrics and modeling merits. Enhancement of scale communality, Reconstructing and normalizing the construct distribution and increases the given model’s efficiency to define the latent construct are some of psychometric merits of parceling. Modeling-Related advantages of Parceling is Estimation stability. Item-based solutions are often unstable and take more iterations to converge, yielding relatively large standard errors of the measurement-level parameters. One consequence stemming from these problems is estimation instability. That is, parameters estimated in item-based solutions may vary substantially even when only small changes are made to the model, making it difficult for researchers to generalize findings. The other aim was to obtain a relatively simple or parsimonious model. 

3. Result

1. In the result section of the manuscript, the authors stated that “The correlation analysis of all the domains of HRQoL and the global HRQoL done using spearman’s rank correlation coefficient.” However, the Overall HRQoL, and its four domains are obtained by taking the average of the corresponding indicator variables and these leads to the overall HRQoL and its domains have a continuous nature. My question is why you compute spearman’s rank correlation coefficient to assess the correlation between the continues variables?

Furthermore the authors does not mention these results in the method part of the manuscript. Your results should be follow the methodology of your study. 

Authors’ response: Thank you for the concern. As you know the Spearman’s rank correlation is be used in two specific scenarios: when working with ranked data and when one or more extreme outliers are present. It measures the strength and direction of association between two ranked variables and as you know WHOQoL items has ordinal nature (Since continuous methods can be used when a variable has four or more levels, the measurement model were analyzed considering the items as a continuous variables). We used spearman’s rank correlation coefficient due to the presence of outliers in our working data because when extreme outliers are present in a dataset, Pearson’s correlation coefficient is highly affected. Therefore, when there is extreme values the spearman correlation coefficient is appropriate that is why we used in our case.

2. The authors also stated, “The final model was relatively simplify and best fitted model.” You mean that your model is correctly estimate parameters of the model?

Authors’ response: Thank you for the comment. It is to mean the model is parsimonious, better fits the data. In statistical modelling, there is no correct model, all models are wrong but we have to find the model that approximate to the truth. Therefore, in our study we have fitted a model with different parameter combination and structure, and compared them with model fit indices. Finally, all the model fit indices of the reported model were satisfactory 

3. The authors stated that “The unstandardized regression coefficients in table 5 represented the amount of change in the……”. If you used unstandardized regression coefficients you are not allowed to compare the coefficients with each other’s, to do so you have to report standardized regression coefficients .

Authors’ response: Thank you reviewer for the comment. We have addressed your comment in the supporting information. We have reported both standardized and unstandardized coefficients. In SEM, it is often advised to report both unstandardized and standardized coefficients, because they present different and mutually exclusive information. Unstandardized coefficients contain information about both the variance and the mean, and thus are essential for prediction. (See the revised manuscript)

4. The authors also reports direct, indirect, and total effect of the variables, during interpreting the total effect of the variables on the outcome variables, the two variables (the one have direct effect t and indirect effect through mediating variable then have total effect) may not have the same unit of measurement. Therefore how to interpret the total effect for such scenarios?

Authors’ response: Thank you reviewer for the concern. In this study, we have considered depression and the four domains of HRQoL as mediator variables and therefore, a variable can have direct, indirect through mediating variables and total effects then we found coefficients for each effects of the variables. Then we have interpreted the coefficients based on their direction and magnitude of association. So the interpretation is straight forward by looking the magnitude and the sign of the coefficients because the unit of measurements were the same. 

5. Finally, the way of presentation of the result is not good, the table the author used is not readable, and to increase the quality of the manuscript clearly present your results.

Authors’ response: Thank you for the comments. We have modified based on your comments, please see the revised manuscript. 

4: Discussion

1. In the recommendation part of the manuscript some of the statements are written multiple times therefore the authors suggested to avoid writing sentences having the same idea over the manuscript. 

Authors’ response: Thank you reviewer for the comment. We have addressed it.

2. Generally, the way you discuss the significant variable was completely not good. The justification the author give for the significant variable was very raw. Furthermore, as I said in measurement part of the manuscript, most of your explanation were not related to what you measured (different from the tool). These makes you result unreliable and hard to comment on it. 

Authors’ response: Thank you reviewer for the comment. We have addressed them.

5: Conclusion 

Generally, the study was done in single site the author could not have to make generalize for the whole population. 

Authors’ response: We thank you for the comment. You are right we cannot extrapolate this results to the whole population because we have conducted this research among prisoners and therefore we can generalize to all prisoners in the study area but we can’t generalize for prisoners out of the study area.

---

## [Editor Report · Decision Letter 2]

26 Jul 2023

PONE-D-22-30782R2Health-related quality of life and associated factors among prisoners in Gondar city prison, Northwest Ethiopia: using Structural Equation ModelingPLOS ONE

Dear Dr. Seifu,

Thank you for submitting your manuscript to PLOS ONE. After careful consideration, we feel that it has merit but does not fully meet PLOS ONE’s publication criteria as it currently stands. Therefore, we invite you to submit a revised version of the manuscript that addresses the points raised during the review process.

In relation to the items raised by the reviewer, please address the following points before more consideration:

- The ± sign was not used anymore to describe SD or SE and you could present the level of variation between parentheses.  

- Please present  Ethiopia as a keyword.

- Please summarize the background section of the abstract.

- Use the same number of decimals throughout the text.

- Present "DV: Physical health" in Table 5 in legend.

- Instead of a null and general sentence at the beginning of the discussion section as "This study investigated the HRQoL in a prison population and explored the relationship between HRQoL and a set of individual socio-demographic, health-related characteristics and  characteristics of detention." present your result.

We look forward to receiving your revised manuscript.

Kind regards,

Kamal Gholipour, PhD

Academic Editor

PLOS ONE
---

## [Author Response · Author response to Decision Letter 2]

1 Aug 2023

Editor Comments:

1. The ± sign was not used anymore to describe SD or SE and you could present the level of variation between parentheses. 

Authors’ response: Thank you dear Editor for the comment. We have accepted your comment and reported the point estimate with the levels of variation within parenthesis. We were reporting the SD/SE to easily show the level of deviation from the point estimate (See the revised manuscript).

2. Please present Ethiopia as a keyword.

Authors’ response: Thank you dear Editor for the comment. We have included Ethiopia as a keyword (See the revised manuscript).

3. Please, summarize the background section of the abstract.

Authors’ response: Thank you dear editor for the comment. We have extensively summarized the background section of the abstract and presented in short and precise manner (See the revised manuscript).. 

4. Use the same number of decimals throughout the text.

Authors’ response: Thank you Editor for the comment. We have made the number of decimal places consistent throughout the manuscript except the regression coefficient. As you can see the regression coefficient, the estimates are too small and if we round these into two decimals they will become zero, which is non-informative. So, we are forced to keep the regression coefficents as it is. If these can’t convince you, we are ready to round the regression coefficients too (See the revised manuscript). 

5. Present "DV: Physical health" in Table 5 in legend.

Authors’ response: Thank you dear Editor for the comment. As you know in this study we have fitted a multivariate model (SEM), that means we have used more than one dependent variables. In our case we have 6 dependent variables such as physical health, psychological health, social relationship, environmental health, overall HRQoL and depression. Therefore, DV stands for the abovementioned dependent variables (we used DV: physical health, DV: psychological health, DV: social relationships, DV: environmental health, DV: overall HRQoL and DV: depression in table 5), To indicate the respective dependent variable for each independent variable coefficients in the final model. 

6. Instead of a null and general sentence at the beginning of the discussion section as "This study investigated the HRQoL in a prison population and explored the relationship between HRQoL and a set of individual socio-demographic, health-related characteristics and characteristics of detention." present your result.

Authors’ response: Thank you editor. We have rechecked thoroughly the discussion and edited as per your comment. (See the revised manuscript)

---

## [Editor Report · Decision Letter 3]

14 Aug 2023

Health-related quality of life and associated factors among prisoners in Gondar city prison, Northwest Ethiopia: using Structural Equation Modeling

PONE-D-22-30782R3

Dear Dr. Seifu,

We’re pleased to inform you that your manuscript has been judged scientifically suitable for publication and will be formally accepted for publication once it meets all outstanding technical requirements.

Kind regards,

Kamal Gholipour, PhD

Academic Editor

PLOS ONE
---

## [Editor Report · Acceptance letter]

18 Aug 2023

PONE-D-22-30782R3 

Health-related quality of life and associated factors among prisoners in Gondar city prison, Northwest Ethiopia: using Structural Equation Modeling   

Dear Dr. Seifu:

I'm pleased to inform you that your manuscript has been deemed suitable for publication in PLOS ONE. Congratulations! Your manuscript is now with our production department. 

Kind regards, 

on behalf of

Dr. Kamal Gholipour 

Academic Editor

PLOS ONE